# EnvSLAM: Combining SLAM Systems and Neural Networks to Improve the Environment Fusion in AR Applications

Giulia Marchesi [1], Christian Eichhorn [1,*], David A. Plecher [1], Yuta Itoh [2] and Gudrun Klinker [1]

1   Faculty of Computer Science, Technical University of Munich, D-85748 Garching b. München, Germany; giulia.marchesi@tum.de (G.M.); plecher@in.tum.de (D.A.P.); klinker@in.tum.de (G.K.)
2   Tokyo Institute of Technology, Yokohama 226-0026, Japan; yuta.itoh@c.titech.ac.jp
*   Correspondence: christian.eichhorn@tum.de

**Abstract:** Augmented Reality (AR) has increasingly benefited from the use of Simultaneous Localization and Mapping (SLAM) systems. This technology has enabled developers to create AR markerless applications, but lack semantic understanding of their environment. The inclusion of this information would empower AR applications to better react to the surroundings more realistically. To gain semantic knowledge, in recent years, focus has shifted toward fusing SLAM systems with neural networks, giving birth to the field of Semantic SLAM. Building on existing research, this paper aimed to create a SLAM system that generates a 3D map using ORB-SLAM2 and enriches it with semantic knowledge originated from the Fast-SCNN network. The key novelty of our approach is a new method for improving the predictions of neural networks, employed to balance the loss of accuracy introduced by efficient real-time models. Exploiting sensor information provided by a smartphone, GPS coordinates are utilized to query the OpenStreetMap database. The returned information is used to understand which classes are currently absent in the environment, so that they can be removed from the network's prediction with the goal of improving its accuracy. We achieved 87.40% Pixel Accuracy with Fast-SCNN on our custom version of COCO-Stuff and showed an improvement by involving GPS data for our self-made smartphone dataset resulting in 90.24% Pixel Accuracy. Having in mind the use on smartphones, the implementation aimed to find a trade-off between accuracy and efficiency, making the system achieve an unprecedented speed. To this end, the system was carefully designed and a strong focus on lightweight neural networks is also fundamental. This enabled the creation of an above real-time Semantic SLAM system that we called EnvSLAM (Environment SLAM). Our extensive evaluation reveals the efficiency of the system features and the operability in above real-time (48.1 frames per second with an input image resolution of 640 × 360 pixels). Moreover, the GPS integration indicates an effective improvement of the network's prediction accuracy.

**Keywords:** SLAM; semantic segmentation; Semantic SLAM; GPS; Augmented Reality; machine learning; AR games; robotics; autonomous driving

## 1. Introduction

In recent years, Augmented Reality (AR) has become mainstream and a clear set of requirements have been defined to differentiate it from simple virtual overlays as found in movies, such as combination of virtual and real, registration in 3D, and real-time interactivity [1]. However, the geometric knowledge of the world is still insufficient for creating more engaging and interactive AR experiences. To address this gap, we focused on creating a new real-time system that can provide semantic information in an AR application while also being efficient and accurate. At the same time, we investigated a new way of improving neural networks' accuracy by introducing external data.

Traditionally, registration in 3D and the user's pose needed for AR have been obtained with marker tracking [2]. While this approach is precise and fast, it has several disadvantages, such as the fact that these markers affect user experience and that the tracking is

limited to areas where they are visible. This led to the development of markerless tracking [3], which is often based on visual Simultaneous Localization and Mapping (SLAM) [4]. SLAM is the task of constructing and updating a 3D map of an unknown environment, while simultaneously tracking the agent's pose within it. Although the use of SLAM for AR is better than the marker-based alternative in terms of usability, interactivity, and camera pose precision, it requires extensive computations. For this reason, using SLAM-based AR applications in mobile devices poses several challenges. There has always been a strong interest in applying AR to smartphones, with examples in the fields of cultural education, digital entertainment, sports, navigation, teaching, and gaming [5]. In fact, mobile phones are promising devices for handheld AR, as they are widely available, cheap, and they contain other sensors that are useful for AR, such as gyroscopes and accelerometers. Examples of challenges of running SLAM-based AR applications on mobile devices are the rolling shutter effect [6], limited memory, and energy efficiency. Another issue is stable pose estimation, since a smartphone has a high degree of freedom in its motions, leading to strong rotations, motion blur, and camera occlusions.

Another problem is that traditional SLAM methods output a purely geometric map of the environment without knowledge of its semantic features. Such semantic features would help applications to better understand the surroundings, adapt to them, and interact with them more realistically. A solution to this issue can be found in the field of Semantic SLAM [7], which fuses a classical SLAM algorithm with a neural network for semantic segmentation to enrich the reconstructed 3D map with semantic information. This has been enabled by the recent success in the field of deep learning, leading to the development of models with improved accuracy, increased speed, and less power consumption [8].

Despite their advantages, existing Semantic SLAM systems thus far have been too inefficient to run on mobile devices—at least without the use of an external server to perform the most resource-demanding computations [9]. Therefore, in this paper we present a new Semantic SLAM system that was designed for mobile use and based solely on on-device computing. We focused explicitly on finding a suitable trade-off between accuracy and efficiency to create a system that can run on mobile devices and, at the same time, represent the world precisely enough with semantic information. Another key research direction is how external information (in this case, GPS and geographical data) can be employed to improve the prediction of real-time neural networks. As a result, we developed EnvSLAM (Environment-SLAM), an above real-time Semantic SLAM system that, in contrast to previous methods, employs a small and efficient neural network and then exploits GPS data to improve its prediction accuracy.

In the end, with a simple AR application we showed an example of how the real-time, efficient and semantic EnvSLAM can be employed. We strongly believe that this paves the way to new AR applications, not only correctly inserted in the 3D world, but also adapting to changing environments and more realistically interacting with real-world objects, based on their semantic characteristics.

Our contributions are as follows:

- Lightweight setup of Fast-SCNN [10], which is trained by combining standardized datasets and a self-made smartphone dataset to segment outdoor environments.
- A smart GPS integration which makes use of the OpenStreetMap database to improve the prediction accuracy.
- Thread-based optimized ORB-SLAM2 implementation for mobile Augmented Reality, which houses the segmentation and GPS functionality without a noticeable decrease in performance.

The paper is organized as follows. First, we have a look at already existing work regarding SLAM with and without neuronal networks for semantic segmentation. Next, we substantiate our design choices and explain the implementation of the algorithm in detail. The subsequent evaluation (quantitative and qualitative) will show the potential of the system, which is then transferred into an example AR Application. The conclusion of the results and an outlook for future work form the end of the paper.

## 2. Related Work

As the core idea of Semantic SLAM is to fuse traditional SLAM algorithms with neural networks, both areas will be reviewed in the subsequent sections. After that, the combination of SLAM and neural networks for semantic segmentation is reviewed separately. Finally, the use of GPS sensors in SLAM systems or deep learning is discussed.

### 2.1. SLAM Systems

The most important use cases for a SLAM system are in the robotic and automotive industries, such as robot navigation, path planning, and driverless cars. Virtual Reality (VR) and AR, which require accurate and low latency pose estimation, can also benefit from SLAM systems. A great number of SLAM overviews exist, see, e.g., in [11–14].

One of the main distinctions between the different SLAM algorithms is the type of information that is processed from the input frame to estimate the camera pose. Direct methods estimate the 3D map and camera pose based on the intensity values of all pixels in a frame, while indirect methods are based on features extraction and data association, thus also being called feature-based. Examples of state-of-the-art direct systems are LSD-SLAM [15], which represents the reconstructed 3D environment with pose graphs and performs data association using probabilistic semi-dense maps; DTAM [16], which is able to reconstruct in real-time a dense map using a single RGB camera; and DSO [17], which is based on a sliding-window optimization and produces a sparse map where 3D points are represented as inverse depth. On the other hand, PTAM [18] can be considered the base for state-of-the-art indirect methods, as it was the first one to separate tracking and mapping in different and parallel threads, which is currently the standard procedure for efficient and real-time SLAM algorithms. Some state-of-the-art indirect systems are DT-SLAM [19] (with a deferring triangulation technique to avoid introducing outliers in the map) and ORB-SLAM2 [20]. As a final alternative, there are semi-direct methods, such as SVO [21], where the initial guess of the relative camera motion and the feature correspondences, derived from the pixels with a high gradient, are computed in a direct way, while the camera pose refinement is computed with a feature-based reprojection error. The back-end of a SLAM algorithm can be filter-based, for example if it uses a Kalman filter as in MonoSLAM [22], which is the first real-time monocular SLAM approach, or key-based, when it has an optimization-based approach as in ORB-SLAM [23]. Our choice of ORB-SLAM2 is explained in Section 3.1.

### 2.2. Neural Networks for Semantic Segmentation

The second main component of our system is a neural network for semantic segmentation, which is the task of assigning a label to every pixel of an image based on the semantic category that it represents. Its main applications are in computer vision and robotics, such as in autonomous driving, scene understanding, and human–machine interaction. Overviews of the field can be found in [8,24–26]. In this case, we focused on the task of 2D semantic segmentation.

While the idea of semantic segmentation has been introduced more than two decades ago, it was not until the rise of deep neural networks that most of its challenges were solved, improving its efficiency and accuracy [26]. Examples of state-of-the-art networks are DeepLabv3+ [27], for semantic segmentation, and Mask R-CNN [28], for the task of instance segmentation. Mask R-CNN has an accuracy of 40 mAP on MS COCO dataset [29] and it is the standard network used in most Semantic SLAM systems, even if it can only run at 5 fps. It is an extension of Faster R-CNN [30], a network for object detection, adding a small fully convolutional network in order to predict a mask for each object's bounding box.

These networks achieve high accuracy at the cost of larger models and slower computation. They are usually not able to run in real-time. However, practical applications have recently forced the field to start considering memory and computational time in the design of new, improved models, especially when targeting mobile and embedded devices. For this reason, a part of the research field deals with different ways of reduc-

ing the model size of a network and to improve its computation and energy efficiency. The initial approach was to employ network compression [31–33] and quantization [34–36] to make a pre-trained model more efficient. Later, the trend shifted toward designing from scratch lightweight CNNs, improving the efficiency by using less computationally expensive convolutions. This process is based on convolution factorization, where the full convolutional operation is replaced with a factorized version, such as depth wise separable convolutions. This is currently the most common way of designing an efficient network. Examples of state-of-the-art models that belong to this second category are DF-Net [37], LEDNet [38], MobileNetV3 [39], FC-HarDNet [40], and Fast-SCNN [10].

### 2.3. Semantic SLAM

Recent development of deep learning has given birth to learning-based SLAM, as described in [41]. Some tasks that can be solved using deep learning in a SLAM system are depth prediction [42], pose estimation [43], and feature matching [44]. On the other hand, the goal of Semantic SLAM is to enrich the SLAM's geometric map with semantic information, thus providing an high-level understanding of the environment. This method can be extremely useful in robotic applications, not only in task planning and object interaction, but also in AR and autonomous driving.

While Semantic SLAM is relatively new, considerable progress has been made in recent years. The idea was introduced by [45], who merged SLAM and object recognition to introduce in the 3D map new high-level semantic scene components, such as planar regions and objects, along with 3D points. Another example by [46] focused on 3D Dynamic Scene Graphs (DSGs) to map spatio-temporal relations between objects and agents in the environment. In this case, the geometric map was built in a first phase and then the image sequence was processed offline to insert semantic information. Later, Semantic SLAM became online and was boosted by the success of deep learning.

Different Goals and Maps: Nowadays, Semantic SLAM systems differ mainly in the way the semantic information is used and for the type of map they create. Regarding the first option, some methods may use additional information to improve the tracking and perform relocalization [7], while others use it to detect and remove dynamic objects, as in DS-SLAM [47] and DetectSLAM [48], thus obtaining a more robust and accurate camera pose and a more precise reconstructed map. In our case, the system aimed to use semantic information to achieve a more immersive and realistic AR experience.

Regarding the second distinction, there are two main options. The first creates an object-aware map that not only is semantically labeled, but also enables single object instances to be recognized and treated as separate entities. The first approach of this type is constituted by SLAM++ [49], while state-of-the-art examples are Fusion++ [50], MaskFusion [51], and DetectFusion [52]. The second employs semantic segmentation networks to enrich 3D map points with semantic information. This is usually based on (1) frame segmentation to obtain a per-pixel class and (2) 2D-3D label transfer to label 3D points. Our proposed method belongs to the second category.

State-of-the-art systems: The most similar state-of-the-art approaches to EnvSLAM are CNN-SLAM [42], EdgeSLAM [9], and [53]. CNN-SLAM is the first work integrating deep learning into a monocular SLAM, namely, LSD-SLAM. It uses the same network to predict a depth map and semantic segmentation, with keyframes as input. The segmentation is employed to enhance the resulting dense geometric map with semantic knowledge and a semantic label fusion method [54] is used to fuse labels over time. On the other hand, EdgeSLAM is the first Semantic SLAM that can run in real-time on a smartphone, even if it needs an external server that addresses of the most computational expensive tasks (i.e., optimization and segmentation). The employed neural network is Mask R-CNN, while the selected SLAM is ORB-SLAM. In the end, [53] is the approach that is most similar to ours. Its goal is to create a semantic 3D map by combining the point cloud generated by ORB-SLAM2 with the semantic knowledge coming from PSPNet-101 [55–58]. The overall

system can run at only 1.8 Hz, far from real-time, and the limiting factor is the speed of the neural network that is applied to every input frame.

### 2.4. GPS Integration

To the best of our knowledge, the idea of employing GPS in a SLAM system to improve the network's prediction has not yet been explored in the literature. In fact, GPS is traditionally used in SLAM to improve—rather than predict—the localization accuracy and the tracking [59].

The closest approaches to ours are those in [60–62]. In [60], the GPS information is employed to augment the result of the segmentation with information such as streets or building names. This is achieved in an offline fashion and not inserted in a SLAM system. Instead, the system in [61] is an autonomous driving platform that tracks itself using a GPS/IMU setup and a LiDAR sensor. The GPS is also used to retrieve information from OpenStreetMap and use it to help detect moving objects in the scene and track them. Our approach, instead, does not aim at using the GPS to help the SLAM system, but to improve the network prediction accuracy. In [62], the GPS coordinates connected to an image are employed to retrieve information from a database regarding the objects that may be present in that image, using these data as a prior for object detection. Similar to this study, we used GPS information to have a rough idea of what the image may contain, although we were interested in a general environmental description rather than in the specific objects locations in the image.

We can also relate our approach to the work in [53], where GPS is used to associate landmarks from Google Maps in the created semantic map. Our goal is different, as they use the GPS to build a topological map, while we employ it to improve the segmentation result. The advantage of our approach is that we trained a general network that can be used in different outdoor environments and then we limited the output classes to the ones that can be found based on the user position.

### 2.5. Semantic SLAM Systems Comparison

Table 1 compares the proposed EnvSLAM with other state-of-the-art methods. Note that the measurements are taken from the original papers and are therefore evaluated on different machines. For this reason, they are not directly comparable, but they still convey an idea of the achieved systems' performance.

**Table 1.** *OA* stands for object-aware map and *Mono* for monocular, while *server* in the *Resolution* column indicates that an external server performs computations. The *FPS* (frames per second) column contains a tick symbol when the methods run in real-time. Thereby the FPS numbers are directly taken from their corresponding publications and, thus, are measured on different machines. *Resolution* refers to neural networks' input resolution. The last four columns state if the system is based on an (I)ndirect SLAM, if the network is applied only on (K)eyframes, if (L)abel fusion is performed and if the GPS information is used to improve the network output.

| Name | Camera | OA | FPS | Resolution | Map | I | K | L | GPS |
|---|---|---|---|---|---|---|---|---|---|
| SLAM++ [49] | RGB-D | ✓ | 20 fps | / | Dense | | | | |
| Fusion++ [50] | RGB-D | ✓ | 4–8 fps | 640 × 480 | Dense | | | ✓ | |
| MaskFusion [51] | RGB-D | ✓ | ✓-30 fps if static | 640 × 480 | Dense | | | | |
| DetectFusion [52] | RGB-D | ✓ | 22 fps | 640 × 480 | Dense | | | | |
| SemanticFusion [56] | RGB-D | | 25 fps | 320 × 240 | Dense | | ✓ | ✓ | |
| Nakajima et al. [57] | RGB-D | | ✓-30 fps | 40 × 30 | Dense | | | ✓ | |
| CNN-SLAM [42] | Mono | | ✓-30 fps | 320 × 240 | Dense | | ✓ | ✓ | |
| EdgeSLAM [9] | Mono | | ✓- 30 fps, server | 1280 × 720 | Sparse | ✓ | ✓ | | |
| Li et al. [58] | Mono | | 10 fps | 1392 × 521 | Semi-dense | | ✓ | ✓ | |
| Zhao et al. [53] | Mono | | 1.8 fps | 1392 × 512 | Sparse | ✓ | | ✓ | |
| EnvSLAM (ours) | **Mono** | | ✓-48 fps | 640 × 360 | Sparse | ✓ | ✓ | | ✓ |

First, our system runs above real-time and it is the fastest one, achieving 48.1 fps with an input resolution of 640 × 360. Other methods that achieve real-time operation, such as CNN-SLAM and the method in [57], have a much smaller network input resolution. In addition, if the input resolution is increased to 1280 × 720, our system can run at 18.2 fps, that is still the best result for high-resolution images. EdgeSLAM is faster only because it uses an external server for heavy processing. Because the frame rate information is taken from the original publications, the hardware for the recordings was different. As test system we used a two-year-old multimedia laptop, thus no powerful hardware was involved compared to other setups utilized in previous publications. Furthermore, the latest ARM chips in mobile devices are considered equally powerful to standard PC hardware, if not better in some specific scenarios, this laptop represents a good indicator for the expected performance. Moreover, GPS integration is a unique feature of EnvSLAM. As another characteristic, the network inferences are performed only on keyframes such that the system is more efficient, avoiding performing a forward pass on every input image, as other systems instead do.

As can be seen in Table 1, CNN-SLAM is quite similar to our proposed approach. Note, however, that it can achieve real-time performance (30 fps) only with a low-resolution input of 320 × 240. Moreover, as in our method, the neural network is employed only on keyframes. However, while we build EnvSLAM on the indirect ORB-SLAM2, their approach develops from the direct LSD-SLAM, which is not ideal when working with mobile devices. EdgeSLAM is another similar approach, but it is noteworthy that, while it achieves real-time performance using the slow Mask R-CNN and running it on an external server, we focused on efficient neural networks and avoided to defer computation externally, as this depends on the availability of a good network connection. The most similar system to EnvSLAM is [53], as it is also based on ORB-SLAM2. We also improved upon this method by using an efficient neural network instead of the slower PSPNet-101 [55] and by performing inference only on keyframes. Because of these two design choices, our system runs at 18.2 fps instead of 1.8 fps at a similar input resolution.

## 3. Design Choices

In this section, we present the motivation for the choice of ORB-SLAM2 and Fast Segmentation Convolutional Neural Network (Fast-SCNN) as building blocks of EnvSLAM and summarize the two methods.

### 3.1. ORB-SLAM2

The SLAM employed in the proposed approach is ORB-SLAM2 [20]. This choice was based on the application's requirements: as the goal was a system that can run in real-time on a mobile device, ORB-SLAM2 was chosen as it is indirect and keyframe-based, and it produces a sparse map. The first characteristic is critical because, as mentioned in [17], an indirect method can provide better performance with a mobile device. In fact, as indirect approaches assume geometric noise, they are more robust to the rolling shutter effect or inaccurate camera calibrations. Regarding the second point, it is more efficient to use a system that does not use every image to triangulate new 3D points, but only keyframes. This allows for expansion by performing network inferences only on keyframes, saving time and computational resources. In the end, a sparse map is more lightweight than a dense one, so suitable with low-memory capacity devices.

Traditional on mobile platforms, ARCore by Google or ARKit by Apple is used to profit from hardware and software optimization. While for most cases the advantages in ease of use and performance would surpass potential limitations, running neural networks with these SDKs is not broadly supported. Furthermore, choosing one SDK would limit the platform independence. Our goal is to leverage a performance oriented and platform independent foundation based on C++.

As a state-of-the-art feature-based SLAM system, ORB-SLAM2 can work in real-time on the CPU. One important feature is the hybrid nature of its map, where both 3D

landmarks and a pose graph are saved, for a total of one metric and two topological maps. ORB-SLAM2 is composed of three parallel threads:

- Tracking: it computes the camera pose for every new frame, based on feature matching and bundle adjustment. It also takes care of relocalization and selects new keyframes.
- Local Mapping: it receives the keyframes and inserts new 3D points in the map. It also optimizes the local map.
- Loop Closing: it uses the last processed keyframe to search for loops. If one is detected, the accumulated drift is computed and the map is corrected.

### 3.2. Fast-SCNN

In EnvSLAM, the network with the best speed/accuracy trade-off needs to be employed, avoiding to use the accurate but slow Mask R-CNN, used in various Semantic SLAM systems [9,50,51]. We claim that the efficiency of the neural network employed in the Semantic SLAM has a huge impact on the overall performance and that a careful choice of this component is essential for the development of a fast and efficient system, that can eventually run also on mobile devices. For this purpose, multiple CNNs were trained and compared using different metrics, evaluating both their accuracy and their efficiency.

First, the networks are trained on the COCO-Stuff dataset [64], which is an extension of COCO [29], the largest dataset for semantic segmentation, by adding annotations for *stuff* classes to the existing *thing* classes. It comprises 164,000 images with dense pixel-level annotations. As the test images are not publicly released, 5000 images were taken from the training set and used for testing. The original dataset has 172 classes: 80 *thing* classes, 91 *stuff* classes, and 1 unlabeled class. These were reduced to 11 classes for two reasons: First, the above mentioned networks do not have sufficient capacity to learn a great number of classes with high accuracy. Second, we were interested in a general understanding of the surroundings rather than in learning to recognize objects. Moreover, some classes are fused to create more general categories. In addition, we only focused on outdoor classes, given the use of the GPS sensor in our system. The final classes are shown in Figure.

All networks were trained following the protocol described in the corresponding paper, until a plateau was reached and for a maximum of 50 epochs. After that, they were evaluated and compared based on accuracy and computational complexity metrics. The second group included the inference time for one image (averaged over the 5000 images of the test set), the number of FLOPs, the number of model parameters and the model size. The results are shown in Table 2.

**Table 2.** Bold characters indicate the best performance. P.A. stands for Pixel Accuracy, F.W. for Frequency-Weighted and M for millions.

| Network | P.A. | Per-Class P.A. | F.W. IoU | Mean IoU | F1 Score | Speed | FLOPs | Params | Size |
|---|---|---|---|---|---|---|---|---|---|
| MobileNetV3 [39] | 52.86% | 26.16% | 34.57% | 17.46% | 25.43% | 19.4 ms | 643.5 | 1.07 M | 12.1 MB |
| Fast-SCNN [10] | **68.52%** | 48.25% | 52.55% | 36.96% | 49.91% | **13.9 ms** | **329.9** | 1.14 M | **4.73 MB** |
| FC-HarDNet [40] | 66.31% | 46.48% | 50.22% | 33.91% | 46.45% | 24.4 ms | 1741.4 | 4.1 M | 16.2 MB |
| DF-Net [37] | 68.26% | 49.41% | 52.51% | 36.39% | 48.87% | 16.0 ms | 3221.7 | 21.1 M | 80.5 MB |
| ICNet [63] | 67.19% | 47.57% | 51.19% | 35.41% | 48.13% | 25.6 ms | 6451.1 | 47.5 M | 181 MB |
| LEDNet [38] | 67.83% | **49.95%** | **52.75%** | **37.33%** | **50.57%** | 31.7 ms | 2284.5 | **0.9 M** | 60.2 MB |

From Table 2, we can note how Fast-SCNN is the best network in terms of computational complexity and the second best in terms of accuracy, making it the ideal choice regarding the accuracy/efficiency trade-off. This led it to be employed in EnvSLAM.

Fast-SCNN is an above real-time semantic segmentation model designed for high-resolution images that is suitable for embedded devices with low-memory capacity. It achieves competitive accuracy (68% mean IoU on Cityscapes [65]), while being able to run at 123.5 fps. The basic idea of its architecture design stems from the observation that the typical solution for accurate semantic segmentation networks is an encoder/decoder

structure with skip connections, while many efficient networks are based on a two- or multi-branch architecture. Fast-SCNN merges these two designs and creates the *learning to downsample* module, which shares the computation of the initial layers of a two-branch CNN, as these initial layers extract low-level features such as edges and corners. Moreover, the model employs depthwise separable convolutions, a key building block for many efficient neural networks.

## 4. Implementation

The following sections detail the fusion of ORB-SLAM2 with the semantic segmentation model, the GPS integration feature, and the Fast-SCNN training details.

### 4.1. Fusion of SLAM and Neural Network

To enrich the ORB-SLAM2 3D map with semantic knowledge, Fast-SCNN needs to be integrated in it. Since the goal was to run the system on a mobile device, the implementation was designed to maximize the speed. It followed these two principles:

1. Minimize the number of neural network inferences.
2. Minimize the idle time of a thread waiting for the segmentation result.

The goal is to minimize the time penalty and computational overhead caused by the neural network introduction, with respect to the original ORB-SLAM2. These rules dictate the design of the system workflow, shown in Figure 1. As can be seen, to follow rule (1) the neural network is employed only on keyframes and not on every input image, as they are the only ones which contribute to the insertion of new points in the map. The creation of a new thread, in comparison to a sequential insertion of the network in the local mapping thread, is justified by the rule (2): minimize the time that the local mapping spends waiting for the segmentation result. In this way, the local mapping can perform some operations in parallel with the segmentation thread before having to wait for its result.

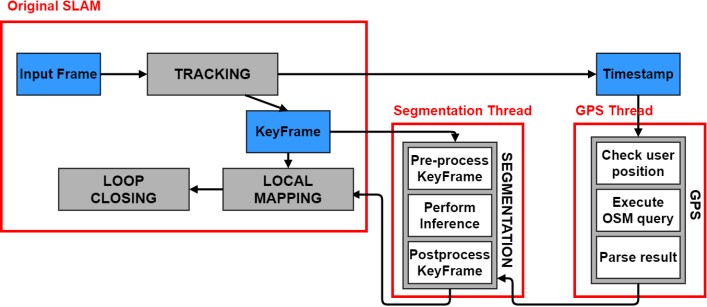

**Figure 1.** EnvSLAM workflow: On the left, a simplified scheme of ORB-SLAM2 is depicted (the complete workflow can be seen in the original paper [20]), while on the right, the additional segmentation and GPS threads, along with their main operations, are visible.

Implementation Details: Compared to the original ORB-SLAM2, a new thread is introduced: the segmentation thread. Its main function is to perform a forward pass on an image using a pre-trained network. Moreover, it takes care of the images pre- and postprocessing and of the visualization of the result. While ORB-SLAM2 runs on the CPU in real-time, the network inference is performed on the GPU. In addition, the design of this thread is highly modular, as a different network can be employed without having to change the implementation.

The system extends the original C++ implementation of ORB-SLAM2 (https://github.com/raulmur/ORB_SLAM2 [accessed on 3 March 2021]). Regarding Fast-SCNN, its training is done using the PyTorch framework, but as its interface language is Python, it needs to be converted to Torch Script with the help of LibTorch, so that it can be loaded and executed in C++. The core part of the segmentation thread is composed by the model loading in C++ and the network inference. During its start up time, the model is loaded on the GPU using the LibTorch API, that deserializes the previously serialized network.

All the subsequent operations are repeated in a loop inside the thread and they are executed as soon as a new keyframe is sent to it. First of all the keyframe is preprocessed: it is converted from BGR to RGB representation and values are normalized to the range [0, 1]. Last, the image is normalized using the mean and the standard deviation values for the COCO-Stuff dataset [64] in order to have a zero mean and a unit variance. After the preprocessing and the network inference, two outputs are obtained: an image with the most probable class label for every pixel and one with the corresponding probability. The two are then sent to the tracking thread, when the system is not initialized, and to the local mapping, during the normal operation, in order to be used during the 3D points creation.

For every 3D point inserted in the map, its class and the corresponding probability are saved. The label is saved only if its probability is higher than a certain threshold (0.5 in our experiments), otherwise the point is considered unlabeled and has the symbolic value 255. This is done to avoid filling the map with outliers by only considering the labels predicted with high confidence. Once the class has been determined at the time of the point creation, it is no longer updated using label fusion methods [42,50], as this would require the reprojection of the already created map points every time a new keyframe is segmented, thus increasing the performed computation.

**Interaction with the Tracking Thread:** The segmentation thread interacts with the tracking thread in two ways. First, during the normal system operations, when the tracking thread chooses a new keyframe and sends it to the segmentation thread to be analyzed by the network. The second interaction happens during the initialization, when the tracking thread is trying to detect the first camera pose and to create an initial map, for whose points it needs the semantic information. To follow rules (1) and (2), a forward pass is executed during the initialization in the parallel segmentation thread only on the current reference frame, and not on all the processed images.

As the biggest interaction between the two threads happens during the initialization, the tracking thread is not slowed down by the network inference during the rest of the operations. Instead, when the tracking is lost and the relocalization is performed, no new keyframes are added, so the segmentation thread is idle, waiting for the tracking to resume.

**Interaction with the Local Mapping Thread:** Once the initialization is concluded, the tracking thread no longer adds new points to the map. During this stage only the local mapping expands the map with new 3D points, so the segmentation thread needs to send its result only to it, as can be seen in Figure 1. When a new keyframe is selected in the tracking, it is sent both to the segmentation and the local mapping threads. They then work in parallel on it before the local mapping must wait for the segmentation image, that is sent from the segmentation thread as soon as it is available. After having received that, the local mapping thread can insert new map points, having information about their class and the corresponding probability.

**Visualization:** A visualization of the created 3D semantic map can be seen in Figure 2. In the original algorithm, the points are colored in red if they are contained in the current local map, otherwise they are black. This is modified in order to color them based on their class (if they are unlabeled, they get marked in black).

**Additional Features:** Two additional concepts were implemented and will be briefly discussed here.

- Label Fusion: Even though a complete label fusion method is not present, we exploited the knowledge of the points' probability to reduce the number of unlabeled points and increase the probability of the labeled ones. In fact, when duplicated points are fused by the local mapping or loop closure threads, we keep the class corresponding to the highest probability between the two.
- Frame Skipping: This is a simple way to improve ORB-SLAM2 efficiency. The idea is based on the fact that it is common for an AR user to not move for some time while using the application. In these moments, the frames are similar to one another, so it is not necessary to process them at a high frequency, but some can be skipped, thus saving computations. Therefore, in the tracking thread, at the end of an iteration,

the frame's pose is inspected. If the user has not moved, the next frame is skipped. This is not performed during the initialization or the relocalization.

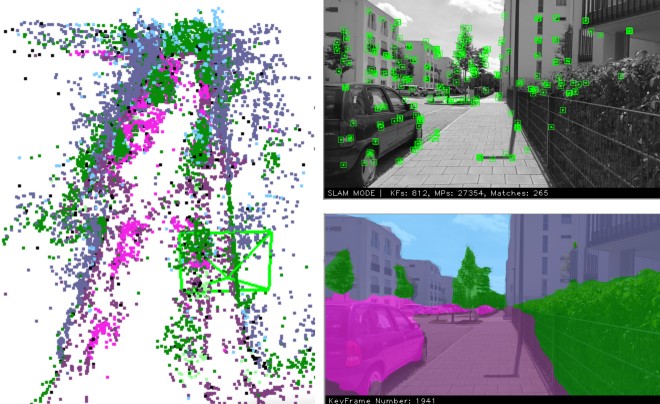

**Figure 2.** It is possible to identify in the 3D maps trees, cars, and buildings, while also seeing the current frame with the tracked keypoints and the semantic segmentation of the last keyframe.

### 4.2. GPS Integration

In this section, we present how the GPS information is integrated into EnvSLAM in order to improve the network predictions. We believe that, as the employed network favors efficiency over accuracy, it may be helpful to aid it with external information. Therefore, the GPS coordinates of the user position are used to understand the surrounding environment and, based on this information, limit the classes in the network output. For example, if from the GPS coordinates the system realizes to be in a park, the classes car and sand can be ignored, so that potentially misclassified pixels are filtered out.

OpenStreetMap: OpenStreetMap (OSM) is an open-source Geographic Information System (GIS), which is a framework that provides the ability to store and analyze geographic data. Its information is stored in topological data structures, associated to some attributes. Here, it was sufficient to work with the *way* elements, which are stored as ordered list of nodes used to represent linear features or areas. During implementation, we experimented with two different types of queries:

1. Circle around the User (*around* keyword): query for all the elements contained in a circle of fixed radius around the user's current location.
2. Camera Field of View (*poly* keyword): retrieve only the features that are present in the user's camera field of view by specifying a triangle representing it.

**Functionalities and Interaction with Other Threads:** First, the GPS integration takes place in a new thread. This is justified by the fact that the OSM query execution is too slow to be part of an existing thread, as the evaluation will show. Moreover, the query needs to be executed at regular intervals to always be updated about the features around the user's current position. This new GPS thread executes the OSM queries, processes their response and takes a decision about which classes are not present in the user's current surroundings. It does not need to be real-time as no threads directly depend on its result.

It needs to interact with the tracking and the segmentation threads, as can be seen in Figure 1. First, the tracking thread needs to communicate the timestamp of the current tracked frame, so that a GPS position can be associated with it. After having found the classes that can be removed, they are communicated asynchronously to the segmentation thread, that will use this information to modify the network output. Note that the segmentation thread does not need to wait at any time for the GPS thread output, it just updates the classes information when it receives it. Finally, while the GPS thread is executing a query, the tracking is going on processing frames. After a query processing, the GPS thread resumes its operation from the current frame, skipping the ones processed in between.

Query Details: In order to query OSM, the Overpass API has been used. A query can be formulated using the offered Overpass QL language and then it can be sent to this API with an HTTP GET request, receiving a response in XML format.

In the GPS thread, a query is executed at the system start up, using the initial user position, the *around* keyword and a radius of 200 m. After this, a new query is executed when the user has moved significantly (50 m of translation or 40 degrees of rotation) with respect to the last executed query. The difference in the two locations is computed comparing the corresponding two GPS coordinates pairs. The distance in meters can be computed using the Pythagoras's theorem. The rotation difference is instead found by computing the difference between the bearings of the two positions, which are the angles between the north direction and the user orientation. During these computations, the user bearings are directly obtained from the rotation vector sensor's values when using our custom dataset. This sensor, or a simple compass as an alternative, is present in all the modern smartphones. Instead, in the Málaga dataset [66], the rotation vector data is not available; therefore, two consecutive GPS values are employed to compute the user's current bearing. While this can lead to inaccuracies, it is not an issue when a mobile device is employed, as the rotation vector can be used instead. Figure 3 shows two examples of OSM queries together with their GPS coordinates and their results.

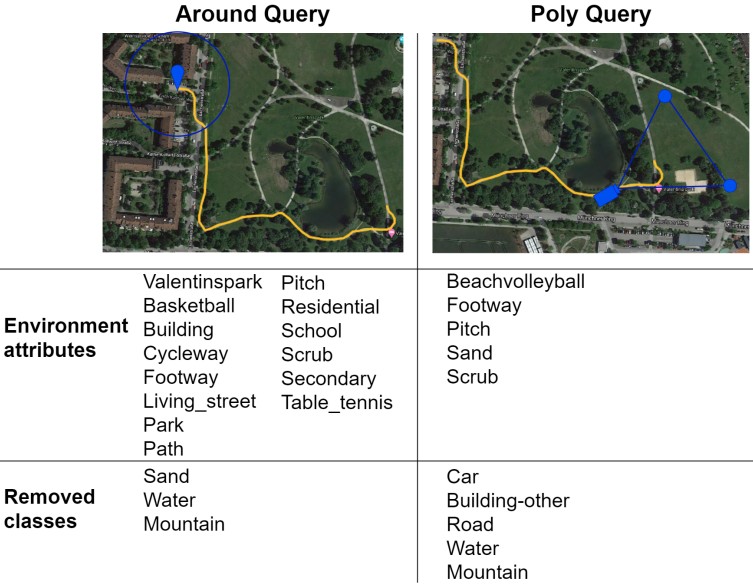

| | Around Query | | Poly Query |
|---|---|---|---|
| **Environment attributes** | Valentinspark<br>Basketball<br>Building<br>Cycleway<br>Footway<br>Living_street<br>Park<br>Path | Pitch<br>Residential<br>School<br>Scrub<br>Secondary<br>Table_tennis | Beachvolleyball<br>Footway<br>Pitch<br>Sand<br>Scrub |
| **Removed classes** | Sand<br>Water<br>Mountain | | Car<br>Building-other<br>Road<br>Water<br>Mountain |

**Figure 3.** The figure shows the coordinates of two Overpass QL queries (the user path is highlighted in yellow). The left image shows an *around* query, while the other is a *poly* query (the camera orientation is shown). The image presents the extracted environment attributes and the corresponding classes that are removed based on those. The attributes are effective in describing the corresponding queried area.

The query response needs to be parsed to extract only the relevant information characterizing the user's surroundings, by choosing the relevant tags. After the parsing, a set of environment attributes is obtained, as can be seen in Figure 3. Based on this set, choices are made regarding which classes to keep in the neural network output and which classes to remove. As only exceptions, the classes *sky* and *tree* are never removed, as they can be found everywhere in an outdoor scenario. Examples of removed classes can be found in Figure 3.

The classes that are found to be removable are communicated to the segmentation thread, that receives this information in an asynchronous way and always uses the most recent result. If the OSM queries fail, due to network delays or temporary server problems, and the last successful query is far from the current user position (150 m of translation or

180° of rotation), no classes are removed in the segmentation thread until a new successful query is executed.

### 4.3. Fast-SCNN Training

Different training protocols and techniques have been compared in order to find the ones that provide the highest training accuracy.

The training was performed on a NVIDIA GeForce GTX 1650 GPU with 4GB, using PyTorch 1.4.0 and CUDA 10.1.243. Given the limited memory of the GPU, the training cycles were bounded in batch size and number of epochs. For this reason, with additional resources a more thorough hyperparameters search could be performed. The employed dataset is COCO-Stuff, reduced to 11 classes as described before.

The employed Fast-SCNN implementation is taken from the open-source version (https://github.com/Tramac/Fast-SCNN-pytorch [accessed on 3 March 2021]). Data augmentation consists of random cropping to $321 \times 321$ resolution, random scaling between 0.5 and 2.0 and horizontal flipping. The training has been done with SGD optimizer, with a weight decay of 0.00004 and momentum of 0.9. The learning rate scheduler is the polynomial one, with update step of 1 and power of 0.9. We employed a batch size of 16 and two auxiliary losses, with weight of 0.4, are also present at the output of the *learning to downsample* and the *global feature extractor* modules.

Transfer learning has been employed using the available pre-trained weights on Cityscapes [65]. To retain the learned features and achieve an higher accuracy, different learning rates are used: a base value of 0.01 in the new output layer and the auxiliary losses' layers, 0.00001 in the initial *learning to downsample* modules and 0.001 in the rest of the network. Moreover, the use of the OHEM (Online Hard Example Mining) loss [67], with 0.7 as the maximum probability and 100,000 as the maximum number of considered pixels, has been found to be beneficial to address the issue of unbalanced datasets. In the end, by analyzing the network gradients and activations, the *dead ReLU* issue has been identified. In order to solve it, substituting all the ReLU activation functions with LeakyReLUs has been found to be beneficial.

Following a common approach [27,39], the training was divided into two stages: the network has been trained completely for 194 epochs and then it has been fine-tuned for other 42 epochs freezing the batch normalization layers, for a total of 236 epochs. After the validation accuracy has reached a plateau, the training has been stopped. As a final note, the original network has been trained for 1000 epochs, so better accuracy can be achieved with more resources.

## 5. Evaluation

The system evaluation has three main components. First of all, the Fast-SCNN evaluation is presented and the GPS integration feature is analyzed in order to understand if it has improved the network prediction. Afterwards, EnvSLAM performance is evaluated from different points of view. Note that the accuracy of the tracking and mapping capabilities of ORB-SLAM2 has not been evaluated, as these aspects have not been improved from the original implementation. We therefore refer to the original paper [20] for more information.

### 5.1. Datasets

While Fast-SCNN has been trained and evaluated on the COCO-Stuff dataset, all the evaluations of EnvSLAM have been carried out using both the public Málaga dataset [66], that represent urban scenarios, and a custom dataset, that instead contains natural environments. The first one is registered from a car, so its videos have more stable movements, while the second is registered on foot with a smartphone, hence its movements are slower and more shaky.

Málaga Dataset: The Málaga Stereo and Laser Urban dataset [66] is registered by a car in the city of Málaga, Spain. From the available measurements we employ the stereo camera (only the left image) and the GPS data (registered at 1 Hz). Timestamps information is also

available for all data, which is essential for synchronization. Specifically, the sequences 6 and 13 of the Málaga dataset were used because of their robust tracking and variate environmental features.

Custom Dataset: To evaluate EnvSLAM a dataset must be registered for two reasons. First, most of the available datasets are recorded by a car or by a robot, while we aimed to develop a system for users of mobile devices. It is therefore necessary to record a dataset on foot and with a mobile device, in order to be slower than a car and with less control than a robot, to test EnvSLAM in real-world scenarios. The second reason is that most of the public datasets are recorded in urban scenarios, since they are aimed for autonomous driving, and thus they lack natural environments, that are instead of interest to us and that are learned to be recognized by Fast-SCNN. In the following, we call this dataset the NP (Natural Phone) dataset.

The dataset has been registered with a Xiaomi Redmi Note 7 smartphone, whose GPS sensor has an accuracy of up to three meters. For acquiring the dataset, an Android application has been developed, which can register videos while also saving the GPS and the rotation vector sensors' values with a frequency of 1 Hz, along with their timestamps. A TXT file is also produced, containing the timestamps of every video's frame.

The dataset has been acquired at 30 frames per second with a 720p resolution (the single frame dimension is $1280 \times 720$). It is composed of 6 sequences, registered in different environments, during different times of the day and with different weather. The videos were registered partially in Germany and partially in Italy, in order to have variations in the environment characteristics and in the quantity of the information available in OpenStreetMap. It is important to note that the dataset presents more challenges with respect to one registered by a robot or by a car. First, a mobile phone's camera has been used, posing for example problems related to distortions caused by its rolling shutter. Furthermore, as it is registered on foot, the sequence is not smooth, but quite shaky and free, and sudden motion is present.

*5.2. Fast-SCNN Evaluation*

Quantitative Evaluation: Different accuracy metrics are evaluated on the 5000 images of the test set of COCO-Stuff and shown in Table 3. The network achieves satisfying results, with 87.4% of pixel accuracy. It also reaches the same mean IoU of the original paper, obtained on Cityscapes (68%). Note that Fast-SCNN can be used both with input images at full and half resolution, without losing too much accuracy and without the need for retraining, as was also proven in the Fast-SCNN paper [10]. In fact, as can be seen in the second row of Table 3, the pixel accuracy at half resolution only decreases from 87.4% to 84.48%.

**Table 3.** Prediction accuracy of the trained Fast-SCNN on the custom version of COCO-Stuff at full and half resolution. P.A. stands for Pixel Accuracy, P.-C. for Per-Class and F.W. for Frequency-Weighted. The results are presented with images at full and half resolution.

| Res | P. A. | P.-C. P.A. | F.W. IoU | Mean IoU | F1 Score |
|-----|-------|-----------|----------|----------|----------|
| Full | 87.40% | 77.29% | 78.28% | 68.0% | 79.81% |
| Half | 84.48% | 74.30% | 74.0% | 62.92% | 75.92% |

Qualitative Evaluation: Some examples of predictions made with our trained Fast-SCNN on the COCO-Stuff test set, along with the ground truths and the original RGB images, can be seen in Figure 4. The network is not able to produce sharp boundaries, but overall it can correctly segment the input images.

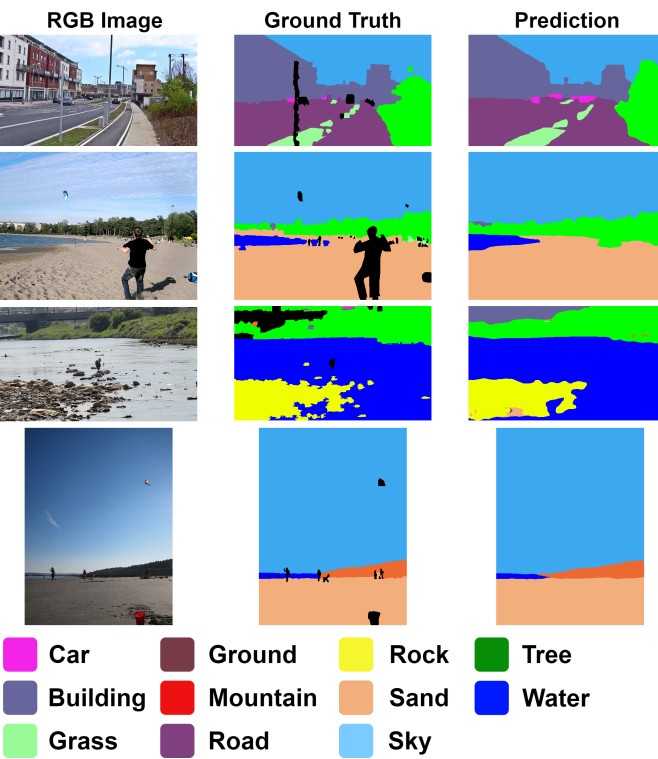

|  | RGB Image | Ground Truth | Prediction |
| --- | --- | --- | --- |

**Car** — **Ground** — **Rock** — **Tree**
**Building** — **Mountain** — **Sand** — **Water**
**Grass** — **Road** — **Sky**

**Figure 4.** Predictions made on the custom version of COCO-Stuff (11 classes). The black pixels in the ground truth images contain classes that were not considered during training.

## 5.3. GPS Integration Evaluation

A novelty of EnvSLAM is the use of the GPS coordinates to improve the Fast-SCNN prediction accuracy. As COCO-Stuff does not have GPS information associated to the images, it is not possible to evaluate the feature using this dataset. For this reason, 15 images from each NP dataset's sequence and 50 from the sequence 13 of the Málaga dataset were randomly chosen and manually annotated with the tool labelme (https://github.com/wkentaro/labelme [accessed on 3 March 2021]).

In the evaluation, the accuracy of the raw network prediction is compared with the one that is obtained when some classes are excluded from the output, given the OSM information obtained with the two different queries previously presented. Both queries are executed with a radius of 200 m around the current user position. Empirically, we found that this value provides enough information to characterize the surroundings, while smaller values do not retrieve enough data from OpenStreetMap and too many classes are then removed. To improve this, an ablation study may be conducted on the value of the radius.

The results are presented in Table 4. Note how the OSM information can improve the network accuracy when the *around* query is used. Even though the improvement is not as expressive as hoped, it is still significant and suggests that including additional information in the network prediction is useful, especially when dealing with a model designed to favor efficiency over accuracy. On the other hand, the information obtained from the *poly* query is not enough to improve the network output, and instead the accuracy is decreased in all the employed metrics. We can conclude that this second type of query, getting information only from the camera field of view area, is not enough for characterizing the environment and therefore more classes than necessary are removed from the network's output. In addition, using the around tag for the query has also other advantages. For example, if the query fails and meanwhile the user is rotating, using the last around query can still provide information about the surroundings, while the poly one is no longer correct because the camera field of view has changed.

**Table 4.** Comparison of the accuracy obtained with the raw network prediction on the NP dataset and the one achieved when introducing OSM information from *around* and *poly* queries. P.A. stands for Pixel Accuracy, P.-C. for Per-Class, and F.W. for Frequency-Weighted. The best results are marked in bold.

| Case | P. A. | P.-C. P.A. | F.W. IoU | Mean IoU | F1 Score |
|------|-------|------------|----------|----------|----------|
| No GPS | 89.94% | 73.92% | **82.42%** | 65.24% | 74.78% |
| Around | 90.24% | **74.07%** | 82.37% | **66.15%** | **76.00%** |
| *Poly* | 89.50% | 72.64% | 81.18% | 64.28% | 74.10% |

An additional evaluation has been conducted in order to inspect the network accuracy in the single dataset's sequences. It showed that the single sequences differ significantly in the accuracy improvement, depending on how much information is available in the queried area from OSM.

Qualitative Evaluation: In Figure 5, some examples are shown to compare the raw Fast-SCNN predictions to the ones obtained when removing classes based on OSM queries. In the first two rows, some wrongly predicted pixels are corrected when removing classes that are known to be absent in the scene. For example, in the first row, the pixels predicted as *water* are mostly correctly predicted when the class is removed, thanks to the additional information from OSM. The same happens in the second row with the pixels wrongly labeled as *mountain*. On the other hand, in the last row, a sample failure of the *poly* is shown: insufficient information is retrieved from OSM, too many classes are removed from the network's output, and the prediction is incorrect.

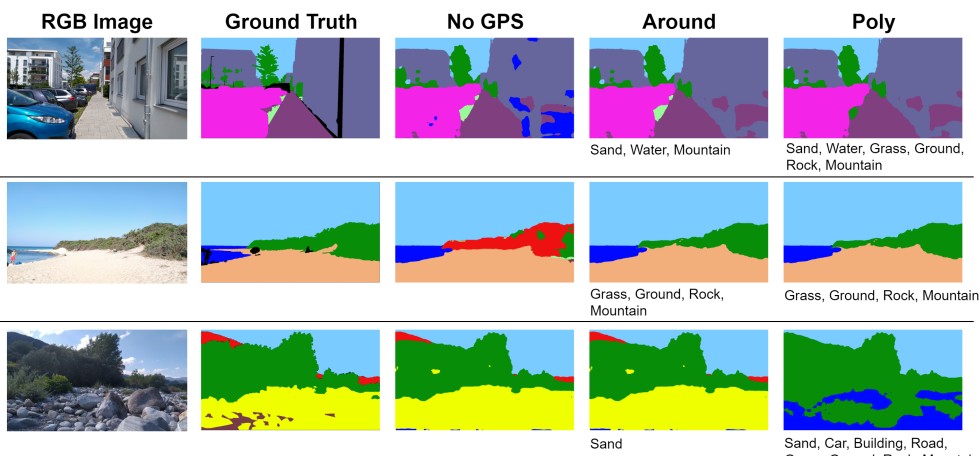

**Figure 5.** Predictions on the NP dataset to compare the raw Fast-SCNN outputs with the segmentations obtained integrating information from OSM. The removed classes are listed below the corresponding predictions.

### 5.4. EnvSLAM Evaluation

It is essential to evaluate the EnvSLAM performance and compare it with the original ORB-SLAM2 to understand how memory, speed, and computational demands have changed with the introduction of the new features. The evaluation has been performed on a laptop with an Intel Core i7-9750H CPU (base frequency 2.60 Ghz) and a NVIDIA GeForce GTX 1650 GPU.

**System Speed:** The first aspect is the number of frames per second that EnvSLAM can work at, which is fundamental for a real-time AR application. Therefore, the frequencies of the original ORB-SLAM2 and of EnvSLAM are compared in Table 5. Note that the speed of the overall system is also the speed at which the main tracking thread runs, while the other threads are not required to strictly run in real-time. The speed is measured twice on all the available dataset sequences, both at full and half resolution, and then averaged.

EnvSLAM has maintained the same frequency of the original ORB-SLAM2, showing that the new features have been introduced in an efficient way. When the system is using full resolution images, it is able to work in close to real-time, at 18.2 fps. More interestingly, with half of the resolution the system can work at more than real-time (48.1 fps), making it the fastest Semantic SLAM system to the best of our knowledge. This efficiency is also because no new operations are introduced in the main tracking thread, that needs to run in real-time, but new threads are created for the additional features. These results make the system suitable for a port to a mobile platform. The bottom part of Table 6 shows the mean execution time of an iteration for the new GPS and segmentation threads, considering only the iterations when computations are performed. Regarding the GPS mean execution time, it remains constant and it is therefore independent from the image input resolution, making this a good feature for a mobile device, where it may be used to get environment information with an overhead that is constant with the input resolution. Instead, the mean execution time of the segmentation thread changes with the input resolution, limiting the use of the system at high resolution.

**Table 5.** Average frames per second at which ORB-SLAM2 and EnvSLAM can work.

| System | Full Resolution | Half Resolution |
|---|---|---|
| ORB-SLAM2 | 18.4 fps | 50.2 fps |
| EnvSLAM | 18.2 fps | 48.1 fps |

**Computational Resources:** The measurements have been executed with the sequence 01 of the NP dataset, both at full and half resolution. First of all, the CPU mean and peak values during the system execution are inspected using Visual Studio profiler. Moreover, the memory (RAM) consumption immediately after the start up and immediately before the end of the system execution has also been measured. The results are presented in Table 6. First, the maximum CPU workload, which happens during the system start up, increases in EnvSLAM. This is probably due to the increased number of threads and variables that need to be initialized during that stage. On the other hand, the average CPU consumption remains constant after the introduction of the new features, proving that the system has been implemented in an efficient way. Regarding the memory, the new system is considerably less efficient. Taking the case of half resolution, ORB-SLAM2 starts with 588 MB and finishes with an occupied memory of 1604 MB, with the memory in between used mostly by the 3D points and the keyframes that are added to the map. On the other hand, in EnvSLAM the required memory is much higher. In particular, the memory consumption between start and end remained constant, but the initial memory allocation is much bigger than with ORB-SLAM2 alone. For this reason, more effort is needed to improve its memory efficiency. Although the memory consumption is high, the system can still fit during its entire execution into 4 GB RAM, as found nowadays in mobile devices.

**Table 6.** This table shows the computational resources that are used by ORB-SLAM2 and EnvSLAM (both at full and half resolution). The last two rows show the mean execution time of the GPS and segmentation threads.

| Resource | ORB-SLAM2 Full | EnvSLAM Full | ORB-SLAM2 Half | EnvSLAM Half |
|---|---|---|---|---|
| CPU Peak | 69% | 88% | 62% | 63% |
| CPU Average | 43% | 43% | 40% | 42% |
| Start Memory | 594 MB | 2132 MB | 588 MB | 2103 MB |
| End Memory | 1392 MB | 2745 MB | 1604 MB | 2933 MB |
| GPS T. | / | 2.0 s | / | 2.4 s |
| Segmentation T. | / | 53.8 ms | / | 23.8 ms |

## 6. Proof of Concept: AR Application

A simple AR application was developed to reveal how EnvSLAM can be employed. It is a desktop C++ application that works in real-time with images of 640 × 360 resolution. Its essential characteristic is the ability to use both the 3D reconstruction and label information in order to adapt to the environment. In particular, the local 3D map is employed to detect a ground plane for a realistic augmentation, while the labels are used to display different 3D models based on the surroundings' features. The application is based on the one available in the original ORB-SLAM2 repository.

The application is developed in a separate thread called viewer thread. As soon as the user clicks on a point of the frame, the respective position is saved to augment there the 3D model. After the click, the currently tracked map points are projected to the 2D image in order to consider only those around the user click and therefore compute the plane around that area. The final set of points is used to fit a 3D plane to the specific map region by performing RANSAC iterations. This works only when the user is in a big planar region with a lot of keypoints, otherwise the plane cannot be computed. The same set of 3D points is then used to collect information regarding the environment: the most frequent label present in the set is computed and used to choose one between three 3D models.

A last implemented feature is the possibility of scaling the objects. This is a necessary feature as the scale in monocular SLAMs is unknown, so the objects are drawn each time with an arbitrary scale. Figure 6 shows three examples of the AR application. This simple application is a preview of what can be achieved in the field of AR with a semantic SLAM system that works in above real-time. Thinking about games, for example, it is possible to have characters that interact with 3D objects and understand their semantic features: they can climb trees, cross rivers, or jump on cars. Moreover, the game itself can adapt and change its content based on the surrounding environment. The newly introduced Superhuman Sports genre contains highly interactive AR sports games, where the understanding of the surrounding allows virtual objects, e.g., a ball, to react naturally towards a collision with the environment [68]. Not only gaming, but also other AR fields can benefit from the additional knowledge that is efficiently inserted in the SLAM system with a small computational overhead. In the field of cultural heritage, EnvSLAM can improve the augmentation of excavation sites [69] or exhibits in museums [70]. Even AR support for navigation could be revolutionized by having semantic input available to prevent collisions with objects.

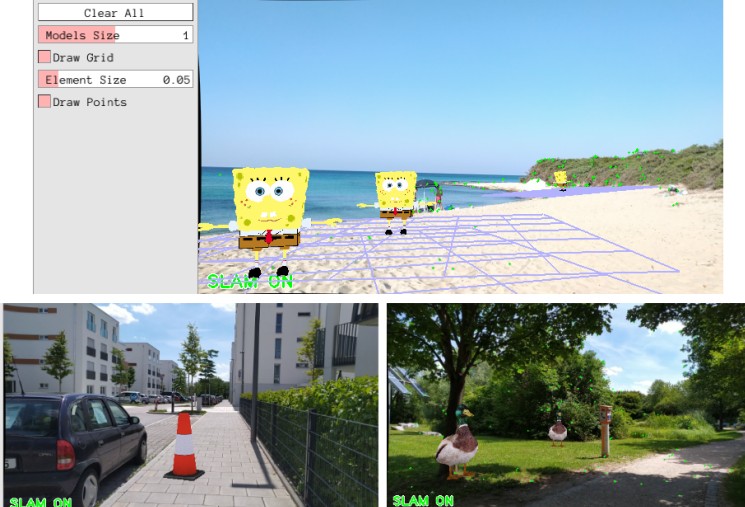

**Figure 6.** The augmented 3D models change based on the surroundings, thanks to the semantic knowledge present in the reconstructed 3D map. If an object is placed on sand, a cartoon character appears, on the pavement a pylon is visible and on grass a duck can be placed. The user has different options: change the model and the plane size, clear all the drawn models, hide the planes, and show the tracked keypoints (green dots).

## 7. Conclusions

We presented EnvSLAM, an innovative Semantic SLAM system that can enrich 3D maps reconstructed by ORB-SLAM2 with semantic knowledge originated from the semantic segmentation network Fast-SCNN. As shown in our evaluation, the developed system can run in above real-time, at a speed of 48.1 frames per second with an input image resolution of $640 \times 360$. To the best of our knowledge, these results make it the fastest Semantic SLAM approach and are therefore promising for a future port to a mobile platform. Such high efficiency was achieved both through an accurate design and a strong focus on an efficient neural network. We also introduced a new way of improving the network accuracy: the GPS integration module. This feature works in real-time during EnvSLAM operations and is used to remove the classes that are absent in the current environment from the network's predictions. The GPS integration resulted in improved network prediction accuracy, which is promising but still highly dependent on the type of performed query and on the quantity and the quality of the OpenStreetMap information in the area of interest.

In the future, EnvSLAM can be employed to expand mobile-based AR applications to adapt to different environments. Virtual characters can interact with their surroundings, allowing the development of new types of rich interactions. Moreover, such a system is useful for autonomous driving or robotic applications, where a real-time geometric and semantic reconstruction of the environment enables improved and safer operations.

## 8. Future Work

EnvSLAM can be expanded in multiple ways. Important aspects will be to optimize the system to run more efficiently on mobile platforms. In a revision we will focus on further interlocking the SLAM algorithm with semantic and GPS data to harvest their synergistic effects. Our future goals are as follows.

**Semantic Knowledge:** The semantic information introduced in the SLAM system will be exploited to make the two elements mutually beneficial. For example, the points' labels can be used to help the tracking, by matching features not only based on their visual appearance, but also on their class, as done in [7].

**3D Map:** The map reconstructed by EnvSLAM can be improved and made more informative: by substituting the semantic segmentation network with one for instance segmentation, the map can acquire knowledge of single objects, as achieved in [46,51,52]. Alternatively, its accuracy can be improved by label fusion techniques, as is done in [42].

**GPS:** The introduced GPS coordinates can be exploited in different ways. In fact, as proven in the evaluation section, the GPS integration is able to improve the network's prediction accuracy only in certain conditions. This shows the need for an improved GPS integration, by considering past queries or by creating more complex relationships between OSM and the removed classes (for instance, execute an around query and then give higher priority to the data contained in the camera field of view). The GPS can also be employed to enhance the system relocalization and loop closure abilities by saving with every keyframe the corresponding GPS coordinates.

**Port to Mobile Platform:** Regarding the system optimization, the memory and resource consumption should be optimized, especially of the ORB-SLAM2 algorithm (see Table 6). Achieving that, a ported version of the system to, e.g., the Unity platform, can bring EnvSLAM on mobile devices. Moreover, an interesting future direction to be able to work with smartphone cameras reliably, would be to use an hybrid approach as the one in [71], where a smartphone camera localization is combined with IMU-based localization to obtain a robust indoor hybrid localization system.

**Author Contributions:** Conceptualization, Christian Eichhorn and Giulia Marchesi; software, Giulia Marchesi; writing—original draft preparation, Christian Eichhorn and Giulia Marchesi; writing–review and editing, Christian Eichhorn and David A. Plecher; supervision, Christian Eichhorn, David A. Plecher, Yuta Itoh, Gudrun Klinker. All authors have read and agreed to the published version of the manuscript.

**Funding:** This research received no external funding.

**Institutional Review Board Statement:** Not applicable.

**Informed Consent Statement:** Not applicable.

**Data Availability Statement:** https://gitlab.lrz.de/ga27let/SemanticSLAM.

**Conflicts of Interest:** The authors declare no conflict of interest.

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
