# Peer review of "EnvSLAM: Combining SLAM Systems and Neural Networks to Improve the Environment Fusion in AR Applications"

_ijgi, doi:10.3390/ijgi10110772_

Round 1
Reviewer 1 Report
In the case of the experiment, a high-performance desktop PC was used, but in the abstract, you said that your target device is smartphone and it seems your testing does not match with this.
It is common to write captions for tables and figures, and to give detailed explanations in the main text. However, most tables and figures do not have captions, but have explanations. Details of tables and figures should be explained in the text.
In general, AR using GPS assumes a mobile environment and a development environment for a smartphone rather than a PC. However, in the case of segmentation with NN that was presented in this text, high-performance PC and high power are required. So, it is questionable whether this can be implemented with EnvSLAM at Smartphone.
Author Response
Please see "Reviewer 1.pdf"

Reviewer 2 Report
Dear Authors,
Please find the attached file for my comments and please update the paper based on the comments.
Best Regards.

Author Response
Please see "Reviewer 2.pdf".

Reviewer 3 Report
This article is well written and present a SLAM system combined with a segmentation and semantic enhancement. The component of work(ORB-SLAM2 and Semantic SLAM) are open source codes and the uniqueness is adaption to mobile devices and integration of GPS signal based semantics extracted from Open Street Map. This work is a nice demonstration that AI can be run on edge devices and it can be used in multiple applications for real time detection of objects. It is well documented the work and it may be useful for other people to reproduce the Natural Phone results. The method is computationally intensive but can be carried out on a phone.
For results in Table 4, PA is PC PA I would have expected PC PA to be larger than PA.Can the author clarify why PA is larger than PC PA?
Author Response
Please see "Reviewer 3.pdf"

Round 2
Reviewer 2 Report
Dear Authors,
Thank you for addressing all my comments and I don't have any further concerns. The paper is accepted from my side.
Best Regards